# Environmental non-governmental organizations and air-pollution governance: Empirical evidence from OECD countries

**Guangqin Li[1], Qiao He[2], Dongmei Wang[3], Bofan Liu[4]***

**1** School of International Trade and Economics, Anhui University of Finance and Economics, Bengbu, Anhui, PR China, **2** Business School, Sichuan International Studies University, Chongqing, PR China, **3** Business School, Northwest Normal University, Lanzhou, Gansu, PR China, **4** School of Government, Sun Yat-Sen University, Guangzhou, PR China

* liubf7@mail.sysu.edu.cn

## Abstract

Based on the panel data of environmental Non-Governmental Organizations (ENGOs) and air pollution in OECD countries, this paper uses econometric model to investigate the governance effect of ENGOs on air pollution. The results show that: ENGOs have a positive impact on the improvement of environmental quality, and the results are still valid after a series of robustness tests; Further mechanism analysis found that the environmental improvement by ENGOs is mainly achieved by increasing investment in environmental protection. This study provides empirical evidence for the effect of ENGOs on air pollution, and further provides ideas for environmental governance.

## 1. Introduction

In the process of industrialization, environmental pollution and its governance have become a global concern. With deteriorating environmental pollution, ENGOs have gradually developed into an important force in promoting environmental governance in countries around the world. From the history of developed countries, ENGOs have gone through the processes of environmental movement, litigation and governance. While it is no longer a major issue in developed counties in the west, environmental pollution still remains their main concern, as the environmental quality is not as good as it used to be. Whether there is a correlation between ENGOs and environmental quality is a topic worth in-depth study.

The U-shaped relationship (Environmental Kuznets Curve) between per capita income and pollutants has been confirmed by a number of empirical studies [1–3]. A review of EKC studies summarized that regulation is the dominant factor to illustrate the decline in pollution when countries grow beyond middle-income phase [3]. Meanwhile, Grossman & Krueger [1] captioned that policy *driven by vigilance and advocacy* plays an important mediating role in the relationship between per capita income and various pollution indicators. Hence, Neumayer & Perkins [4] concluded that any factors that have significant power on environmental policy could affect the environmental outcomes.

**Funding:** This work was supported by the Research Project of Anhui University of Finance and Economics (Green Development Effect of China's Green Credit Policy, Recipient GL), the Research Project of Sun Yat-sen University (13000-31610513, Recipient: BL), Shandong natural science foundation project (ZR2020QG043), and the social science foundation project of Shandong Province (20DJJJ033, Recipient: Not applicable).

**Competing interests:** The authors have declared that no competing interests exist.

Among other factors, ENGOs are the primary driving force for vigilance and advocacy of the environmental policy [2]. ENGOs started to take part in environmental governance since 50's last century [5]. Since then, ENGOs have been transformed from a pure organizer of environmental movements to a key player in environmental litigation and governance. For example, it played a crucial role in the passing and enforcement of the *Clean Air Act in the United States* [6].

Despite the outstanding contribution of ENGOs to environmental protection, literatures in discussing their effect on environmental outcomes are relatively rare. The main focus of current literatures is how ENGOs can influence individuals, enterprises and government behaviors. For example, some studies found that ENGOs can encourage the public to act upon environmental issues through education [7]. Furthermore, ENGOs push enterprises to take greener behavior through such actions as sending complaint letter, organizing mass protest and lodging civil litigation [8,9]. Moreover, ENGOs in corporation with a company's stakeholders have strong impact on its environmental behavior [10–12]. Then, there are plenty of studies on the impact of ENGOs on government environmental behavior, most of which investigated from a qualitative perspective about how ENGOs can affect environmental quality of a country or region [4,10,11], and how they put pressure on policymakers to protect the environment together with their international counterparts [13].

The majority of current literature on ENGOs and environmental governance is categorized to public management, with qualitative analysis or case study being a main research direction. From the perspective of environmental economics, the current emphasis on the participation of ENGOs in environmental governance is insufficient [14], especially on quantitative research, which can hardly match the development of ENGOs and their role in environmental governance. Therefore, this study uses the cross-border panel data of OECD countries to investigate the impact of ENGOs on the atmospheric environment quality of these countries, so as to bridge the gap of ENGOs' effect on environmental outcomes.

The rest of the article is arranged as follows: Section 2 of this paper reviews the related literature and provides our hypothesis. Section 3 describes the research design and the following section tests robustness of our model. In section 4, we employ regression analysis to investigate how ENGOs affects various pollution indicators. The last section presents the conclusions and provides some policy implications.

## 2. Literature review and theoretical hypothesis

### 2.1 Literature reviews

To sum up, there are several ways for ENGOs to participate in Environmental Governance: first, ENGOs can raise their environmental awareness by holding environmental campaigns [15]; second, if their domestic ENGOs are relatively weak, they can improve their environmental discourse right by cooperating with international ENGOs in other developed countries [13]; third, in order to better participate in environmental governance, ENGOs take the initiative to establish interactive relations with enterprises and the government, so as to promote the realization of the normal functions of ENGOs [16,17]; fourth, ENGOs cooperate with media organizations to transmit environmental information to the public through the media, and improve the public's environmental awareness [18]; fifth, for some countries with relatively small land area, environmental governance needs transnational cooperation, and ENGOs also need to cooperate closely with ENGOs in other countries to jointly deal with cross-border pollution problems [16,19–21]; Sixth, ENGOs regularly publish reports on the environmental behavior of the government or enterprises to the public through information disclosure, so as to promote the government and enterprises to adopt more green

environmental behavior [22–25]. With the growth of ENGOs, the fields of different ENGOs are constantly subdivided, and the ways to ENGOs participate in environmental governance are constantly changing.

## 2.2 Theoretical hypothesis

What is the relationship between ENGOs and environmental quality? The following is an analysis of the development of ENGOs in the UK, the United States, Germany and South Korea, which have experienced environmental pollution problems in the process of industrialization.

The United Kingdom, the first industrialized country, was most severely affected by environmental pollution, as evidenced by the smog incident in London in the 1950s [26]. ENGOs that rise with environmental pollution have developed rapidly in the UK, going through three stages characterized by elitism, populism and strategicism, and played the role of *a third-party manager* in making up for the failure of the state and the market [27]. After more than a century of development, the number of ENGOs registered in the UK has reached 200 according to Hilton et al. [28].

In the United States, ENGOs are a substantial environmental force outside all levels of government and scientific research institutions [29]. As a principal organ of environmental protection movements, they lodge lawsuits to court, appeal to parliament and lobby for specific environmental issues, and eventually realize management, compensation, supervision and control over pollution and ecological destruction through legislation. As the largest ENGO in the US, the World Wildlife Fund (WWF) has 1.2 million members across the country and international branches in dozens of countries, including China [30]. In addition to large organizations, there are also many small ENGOs in the US, such as the Pacific Environment and Resource Center established in 1987.

In the 1950s and 1960s, Germany was eager to emerge from the backwardness of its postwar situation and took a path of *pollution before treatment*, leading to increasingly severe environmental pollution, where industrial waste filled Rhine River and no blue sky could be seen at Ruhr industrial zone [31]. The ecological environment in Germany today, however, has been improved exceptionally, thanks to ENGOs' immense contribution. Germany has over a thousand ENGOs, large and small, with more than two million employees. The largest one is the Nature Conservancy, with a history of over 100 years and about 400,000 members.

South Korea has also experienced serious pollution caused by rapid industrialization that has spawned a group of socially responsible ENGOs, of which the Environmental Movement Alliance is the largest, with branches and 85,000 members in 47 regions. Driven by the organization, South Korea's ENGOs have been growing and played a pivotal role in the environmental struggle from 1988 to 1992 mainly by participating in the environmental campaign against the construction of new nuclear power plants.

Developed countries have experienced the stages from pre-industrialization to industrialization and then post-industrialization; therefore, their environmental governance has correspondingly gone through the stages of pollution before treatment, control while pollution and re-control with less pollution, which ENGOs have been growing in this process. It can be seen from such developed countries as the United Kingdom, the United States, Germany and South Korea that ENGOs have been associated with environmental pollution and become major participants and promoters of environmental pollution control. Therefore, we believe that there is a hypothetical relationship between the development of ENGOs in these countries and the environmental quality of these countries:

**Hypothesis 1**: With the growth of a country's environmental NGOs, the country's environmental pollution will be reduced (environmental quality will be improved).

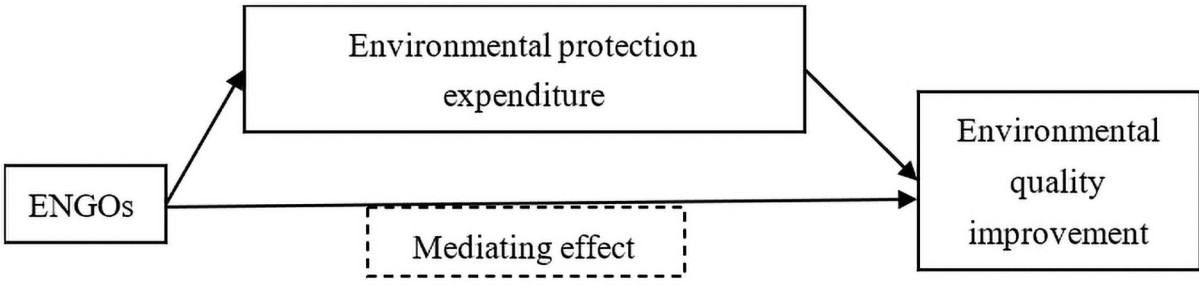

**Fig 1. Impact mechanism of ENGOS on environmental quality improvement.**

Environmental governance can be classified into formal and informal regulation [32,33]. Formal environmental regulation mainly refers to the administrative and economics measures taken by a government [34]. The former includes implementing administrative orders to reduce pollution emissions or the use of chemical energy, restricting entry access for enterprises with certain industrial attributes, and shutting down pollution enterprises compulsorily. The effect of administrative measures is remarkable. The economic measures include taxation to pollutants, where the result is determined by tax severity and the ability of enterprises to bear the tax, or the provision of subsidy to nice and cleaner technology. Informal environmental regulation mainly refers to the way to restrict environmental behavior by improving residents' moral level and environmental awareness [35]. While formal environmental regulation addresses environmental issues directly, informal regulation exerts little immediate effect on environmental governance; instead, it achieves such effect through the influence of other factors to reduce environmental pollution and improve environmental quality. The role of ENGOS is informal regulation in environmental governance [25]. It can be seen from the previous analysis that ENGOS can influence environmental pollution. But how ENGOS affect environmental pollution is still unknown. Fig 1 shows the main ways of how ENGOS influence environmental quality improvement.

As is shown in Fig 1, the traditional mechanism of environmental governance of ENGOS mainly consists of two parts. First, ENGOS ameliorate environmental quality mainly through education and information disclosure [19,25]. As the publicity becoming more aware of environmental problems, they push the government to increase budget on environmental expenditure [7]. In addition, ENGOS meliorate environmental quality by influencing government environmental expenditures (including enterprises' environmental protection expenditure). However, due to data constraints, it is hard to obtain environmental awareness gained by residents through ENGOS' education, but we can estimate the effect of environmental expenditure on ENGOS' improvement of environmental quality. Based on this, we propose the following hypothesis:

**Hypothesis 2**: ENGOS ameliorate environmental quality by means of increasing environmental expenditure, and further accelerate life expectancy and birth rate of the birth rate of the whole society.

## 3. Methodology and data

### 3.1. Model specification

According to Hypothesis 1, we construct the following econometric model to investigate the impact of ENGOS on environmental pollution emissions:

$$Pollutant_{it} = \alpha_0 + \beta \times engo_{it} + Z \times \lambda + \mu_i + \upsilon_t + \xi_{it} \tag{1}$$

where subscripts *i* and *t* represent country and time respectively; *Pollutant* is the dependent variable, representing the emission of air pollution; *engo* is the key explanatory variable, representing the measurement index of ENGOs; *Z* is other control variables that affect the *Pollutant;* *λ* is the corresponding coefficient matrix of control variables; *μ* and *ν* are regional fixed effect and time fixed effect, respectively, indicating unobservable factors affected by region and time; *ξ* is the random disturbance term; *β* is the coefficient of our concern, where if it is less than zero, it shows that ENGOS have the emission reduction effect of environmental pollution, and the Hypothesis 1 can be verified.

We take the natural log of all the dependent variables to render their distribution less skewed, thus mitigating potential problems with heteroscedasticity. With respects to the explanatory variables, we take the natural log of all cardinal variables, namely income, ENGO and business lobby strength, literacy, population density and the Gini coefficient, to allow an easy understanding of elasticity interpretation. In addition, a log-linear form is also commonly used in the EKC literature and exhibited a better fit with data at hand. As for the concerns of the income variable, given the result from the EKC literature suggesting non-linear effects on population, we include income squared in the model to be estimated. If both the linear and the squared term are insignificant, it is a typical sign that their relationship is more complex and a cubed income term is added to the estimations. Theory does not predict non-linear effects of other explanatory variables, which is why they all enter the estimations in linear form only.

## 3.2. Variables and data

**3.2.1 Key explanatory variables.** According theoretical literature, the strength of ENGOs is based on their financial resources and membership quantity. However, poor availability of these data makes it impossible to conduct any empirical research on them. Therefore, many studies use the number of ENGOs per capita as proxy for measuring the strength of ENGOs (Binder & Neumayer, 2005). Although some argue that the number of ENGOs does not represent the participation of ENGOs in public affairs, it is regarded as the best proxy at present. Therefore, we adopt it to measure the influence of ENGOs in a given country.

The data of numbers of ENGOs comes from the online library of United Nation (website: https://research.un.org/en/ngo), where there is a database of NGOs and Sustainable Development registered with the United Nations (website: http://esango.un.org/civilsociety/1ogin. do) that provides four searching catalogues, i.e., by region, status, field of activity and type of organization. We searched the directory of environment-related NGOs in the database by field of activity, entered the directory homepage of ENGOs (http://esango.un.org/ civilsociety/withOutLogin.do?method=getFieldsOfActivityCode&orgByFieldOfActivity Code=8&orgByFieldOfActivityName=Sustainable%20Development&sessionCheck= false&ngoFlag=), and obtained 16,683 NGOs that meet our requirements and defined them as ENGOs, all of which provided information including organization name, registration address, contact information, main practice areas, activity regions and countries. After sorting out the data, we managed to make a panel database about the numbers of ENGOs in OECD countries from 2000 to 2014. The data of population are derived from World Bank.

Given that we can pinpoint the time of its establishment in the directory of ENGOs, this paper builds a panel database of the number of ENGOs owned by OECD countries by year and by country. Considering the comparative of ENGOs among different countries, we build two indexes, i.e., *engo_pop* and *engo_sq*, where *engo_pop* is defined as the variable of numbers of ENGO per capita while *engo_sq* is defined as the numbers of ENGOs per miles of a country.

Hence, we get the following formula:

$$engo\_pop_{it} = engo_{it}/pop_{it} \tag{2}$$

$$engo\_sq_{it} = engo_{it}/sq_{it} \tag{3}$$

where, $pop_{it}$ and $sq_{it}$ respectively refer to the population and territory of country $i$ at time $t$, using millions of people and ten thousand square kilometers as units. The above two indicators respectively represent the number of ENGOs owned by one million people and the number of ENGOs owned by ten thousand square meters of public land. Taking the United Kingdom as an example, there were 41 national ENGOs registered with the United Nations by the end of 2015, which means that for every 1.7 million people of per 6000 square kilometers, there was one ENGO. Population and area data for OECD countries in a calendar year is from the OECD Statistical Information Network (https://stats.oecd.org/).

**3.2.2 Dependent variable.** There are many indicators that can measure the effectiveness of environmental governance. Among the data provided by the OECD statistical database, the indicators that can represent environmental quality are mainly air pollution indicators, but the statistics of other pollution indicators are not substantial. In this regard, this paper tests five common air pollutants, namely, $PM_{2.5}$, Greenhouse-gas, NO, $NO_2$.

We use the mean concentration of $PM_{2.5}$ as the indicator of haze index, which was calculated in OECD statistical database. Its average maximum concentration in OCED countries is 30.6μg/m$^3$ and the average minimum concentration is 13.6 μg/m$^3$. The overall air quality is good, but large differences exist among regions. Meanwhile, we use the proportion of population exposed to air with $PM_{2.5}$ exceeding 10μg/m$^3$ as the indicator of air pollution intensity. As for the rest pollutants indicators, they are all measured by total emission quantity. We take the form of log in the estimation process to avoid coefficient bias caused by unit differences. All the data for the five pollution indicators comes from OECD database from 2000–2014.

Taking $PM_{2.5}$ as an example, we analyzed the trend of air pollution in OECD countries from 2000 to 2015. In 2000, the 34 OECD countries can be divided into three groups according to $PM_{2.5}$ concentration. The least polluted group includes Canada, Australia, Ireland and Nordic countries, with an average concentration of less than 10 μg/m$^3$; while the number was beyond 20 μg/m$^3$ in the most polluted group with countries like Mexico, Korea and some eastern European countries. The rest countries of OECD belong to the medium polluted group, with a concentration level between 10–20 μg/m$^3$.

Compared with 2000, most of the 34 OECD countries showed a downward trend of haze pollution concentration in 2005. The average concentration of smog pollution in Mexico, Sweden and Chile decreased by over 2 μg/m$^3$, over 1 μg/m$^3$ in Italy, the United States, Greece, Israel, Norway, Germany, South Korea and Belgium decreased, and less than 1μg/m$^3$ in 12 other countries. However, the number rose in 10 other countries, i.e. Hungary by 3.5μg/m$^3$, Slovakia by 2.8μg/m$^3$, and the Czech Republic, Portugal, Austria, Poland and Estonia by 1–2μg/m$^3$. In addition, slight deterioration was observed in Finland, Slovenia and Canada.

Compared with five years ago, smog pollution in Mexico, the United States and South Korea fell further in 2010 by 3.7μg/m$^3$, 2.9μg/m$^3$ and 2.1μg/m$^3$ respectively. Meanwhile, the smog pollution in Slovenia, Austria, Canada, Portugal and Hungary shifted from deterioration to improvement, decreasing by 2.3μg/m$^3$, 2.1μg/m$^3$, 2.0μg/m$^3$, 2.0μg/m$^3$, 0.9μg/m$^3$, respectively. However, smog pollution worsened in nine other countries, including Iceland, Sweden, Luxembourg, the Netherlands, Belgium, Germany, Israel, Greece and Turkey. As a result, the number of countries with reduced smog dropped from 24 to 20. Among them, Israel, Poland, Greece and Turkey saw the steepest deterioration, with the numbers rising by 3.0μg/m$^3$, 3.7μg/

m$^3$, 5.1μg/m$^3$, 7.4μg/m$^3$, respectively. Compared with 2010, the number of countries with ameliorated haze pollution in 2015 fell to 17, accounting for half of the 34 countries. The situation slightly deteriorated by rising less than 1μg/m$^3$ in six other nations, including the United States, New Zealand, Spain, Portugal, Chile and Australia. South Korea saw the most serious smog pollution with the number reaching 5.3μg/m$^3$. Austria, Switzerland, Slovenian, Italy, Mexico and other countries saw the number rising more than 2μg/m$^3$. However, Poland, Turkey and the Netherlands with smog pollution deterioration five years ago became the countries with highest improvement, where the numbers decreased by 7.4μg/m$^3$, 3.2μg/m$^3$ and 3.1μg/m$^3$, respectively.

Overall, air pollution levels in the 34 OECD countries have shown a trend of improvement, but there are still fluctuations in some areas. This is related to the level of economic development, industrial structure changes and other factors in certain years.

**3.2.3 Controlled variables.** Drawing on existing researches [36,37], and combining the availability and comparability of data, we mainly choose the following indicators as control variables.

Manufacturing structure (*Second*): measured by the share of manufacturing value added in GDP. It is used to examine the relative development rate of manufacturing. The higher the proportion of manufacturing, the greater the likelihood of environmental quality deterioration. Its impact coefficient on each environmental pollution variables is expected to be positive.

Manufacturing growth rate (*Secondr*): measured by the real growth rate. It is used to examine the relative growth rate of manufacturing. It can be used to determine the growth rate of the manufacturing comparing with its base period, and to further decide whether it will accelerate or slowdown based on its current status. Given that the accelerated development of a manufacturing is bound to cause environmental degradation, we expect the rate symbol to be positive.

Structure of construction industry (*Build*): measured by the added value of the construction industry in proportion of GDP. It is mainly used to investigate the degree of environmental pollution affected by the construction industry. The higher the proportion of the construction industry, the more dust and garbage will be in the construction process, which can also accelerate the deterioration of environmental quality. In addition, the construction industry may cause environmental quality deterioration, so its impact coefficients of various environmental pollution variables are expected to be positive.

Growth rate of construction industry (*Buildr*): measured by the growth rate in real terms. It is used to examine the relative development rate of the construction industry, as a way to determine whether the growth of the construction industry can lead to the environmental deterioration. Its coefficient is expected to be positive.

Electricity consumption (*Ele*): measured by electricity generation. Because of the instantaneity of electricity production and consumption, electricity generation equals to electricity consumption when dismissing export and import of national electricity. This index is mainly used to investigate the degree of environmental pollution caused by power generation process. Due to the lack of data on the proportion of clean energy, the impact of this variable on environmental pollution is uncertain.

Environmental investment *(Env_i)*: measured by the proportion of environmental expenditure in total expenditure. Financial expenditure related to environmental protection and its share of total fiscal expenditure are decisive in a country's environmental quality improvement. The more a country spends on environmental protection, the more importance it attaches to environmental governance, hence the more secure its environmental quality will be accordingly. Therefore, its coefficient is expected to be positive.

**Table 1. The main variables definitions and descriptions.**

| Variable | Definitions | Obs. | Mean | SD | Min. | Max. |
|---|---|---|---|---|---|---|
| Dependent variable | | | | | | |
| $PM_{2.5}$ | Mean concentration of $PM_{2.5}$ (μg/m$^3$) | 510 | 13.638 | 5.601 | 3 | 30.600 |
| $PM_{2.5}\_pop$ | Proportion of population exposed to air with $PM_{2.5}$ exceeding 10μg/m$^3$ (%) | 510 | 68.854 | 38.893 | 0 | 100 |
| Ln$CO_2$ | Carbon dioxide emissions (kiloton) | 510 | 3.681 | 1.534 | 0.642 | 8.649 |
| Ln$NO$ | Nitrogen oxide emissions (kiloton) | 510 | 5.862 | 1.427 | 3.119 | 9.982 |
| Ln$NO_2$ | Nitrogen dioxide emissions (kiloton) | 510 | 5.134 | 1.855 | 0.233 | 9.600 |
| Key explanatory variable | | | | | | |
| $Engo\_pop$ | The number of ENGOs owned by 10,000 people (ENGO/head count) | 510 | 1.314 | 3.297 | 0.003 | 20.628 |
| $engo\_sq$ | The number of ENGOs per 10,000 square kilometers (ENGO/national territorial area) | 510 | 1.505 | 5.854 | 0.001 | 33.803 |
| Control variable | | | | | | |
| $Secondr$ | Real growth rate of manufacturing (%) | 510 | 2.066 | 5.755 | -20.887 | 19.424 |
| $Second$ | The ratio of manufacturing value added to GDP (%) | 510 | 22.167 | 5.847 | 6.787 | 39.483 |
| $Ele$ | Consumption of electric power ($10^{16}$watt) | 510 | 30.846 | 71.818 | 0.042 | 440.000 |
| $Env\_I$ | The ratio of environmental expenditure to total fiscal expenditure (%) | 510 | 0.971 | 0.896 | -1.840 | 5.334 |
| $Buildr$ | The real growth rate of construction industry (%) | 510 | 1.208 | 8.340 | -41.005 | 51.133 |
| $Build$ | The ratio of construction value added to GDP (%) | 510 | 6.178 | 1.554 | 1.652 | 11.702 |

Data sources: authors' calculation and OECD Statistical Information Network(https://stats.oecd.org/).

Woo et al. [37] also controlled variables such as FDI and technological progress in their research. Because the indicators of FDI and technological progress are incomplete in the time of this study, these factors are not taken into account. Due to the relatively substantial data in 2000 and the late accession of Lithuania to the OECD, this paper does not consider Lithuania for the time being. In addition, many data from the OECD Statistical Information Network after 2015 is not complete, so we selected the data of the 34 OECD countries from 2000 to 2014 as our sample. Table 1 provides descriptive statistics of the main variables.

## 4. Results and discussion

### 4.1. PM$_{2.5}$ emission reduction effect of ENGOs

Table 2 reports the estimation result of ENGOs on $PM_{2.5}$ emission reduction effect. Column (1) controls the region fixed effect, but not the year fixed effect, while column (2) controls the year fixed effect, but not the region fixed effect. The estimation results show that the value of goodness-of-fit rises from 0.028 to 0.059, suggesting that the interpretation ability of the model is dramatically enhanced, after controlling year fixed effect. However, the change in the coefficient of ENGOs is far from obvious, with the number being -0.287 and -0.291, both significantly negative at the level of 1% beyond. It shows that the impact of ENGOs on PM$_{2.5}$ is relatively stable and hardly affected by the fixed effect of region or time. Therefore, we can infer that the development level of ENGOs in a country has a notable negative impact on its average $PM_{2.5}$ concentrations; that is, it has the effect of mitigating environmental pollution. More precisely, with an increase of 1 unit in the number of ENGOs per million people, the average $PM_{2.5}$ concentration value drops by 0.29 unit. Hence, hypothesis 1 is verified.

The estimation results in columns (3) and (4) show that the coefficients of ENGOs are notably negative at the level of 5% beyond, being -0.162 and -0.176 respectively, indicating that there is no noticeable change in the coefficients after controlling year fixed effect. However, a conspicuous change exists when comparing the coefficients from columns (1) and (2),

**Table 2. PM$_{2.5}$ emission reduction effect of ENGOs.**

| Explanatory variable | Dependent variable: PM$_{2.5}$ | | | | Dependent variable: PM$_{2.5}$_pop | |
|---|---|---|---|---|---|---|
| | **(1)** | **(2)** | **(3)** | **(4)** | **(5)** | **(6)** |
| *engo_pop* | -0.287*** | -0.291*** | -0.162** | -0.176** | -1.955*** | -2.967*** |
| | (0.063) | (0.062) | (0.079) | (0.080) | (0.588) | (0.740) |
| *Secondr* | | | 0.103** | | | -0.108 |
| | | | (0.052) | | | (0.082) |
| *Second* | | | 0.196*** | | | 0.742** |
| | | | (0.059) | | | (0.323) |
| *Ele* | | | -0.002 | | | -0.466*** |
| | | | (0.002) | | | (0.126) |
| *Env_I* | | | -0.629** | | | 1.583 |
| | | | (0.244) | | | (2.022) |
| *Buildr* | | | -0.031 | | | -0.063 |
| | | | (0.035) | | | (0.053) |
| *Build* | | | 0.072 | | | 0.801* |
| | | | (0.167) | | | (0.455) |
| Constant | 13.015*** | 13.769*** | 9.550*** | 9.884*** | -13.819* | -1.891 |
| | (0.259) | (0.942) | (1.577) | (1.917) | (8.138) | (8.712) |
| *N* | 510 | 510 | 510 | 510 | 510 | 510 |
| Time-fixed | N | Y | Y | N | Y | Y |
| Region-fixed | Y | N | Y | Y | N | Y |
| R-squared | 0.028 | 0.059 | 0.094 | 0.121 | 0.947 | 0.952 |
| F test | 20.859 | 2.615 | 6.468 | 2.731 | 2459.536 | 746.015 |
| | [0.000] | [0.000] | [0.000] | [0.000] | [0.000] | [0.000] |

*Notes*: The numbers in parenthesis are robust standard errors; P value are in square bracket;

*, **, and *** represent 10%, 5%, and 1% significant level, respectively.

indicating that environmental non-governmental organizations may affect *PM$_{2.5}$* through other control variables. Among the control variables, the growth rate of manufacturing and the proportion of manufacturing in GDP are both distinctly positive, marking that the development of manufacturing industry has indeed increased its average concentration of *PM$_{2.5}$*. Yet, the growth rate and proportion of construction industry have no noticeable impact on *PM$_{2.5}$*, which can be attributed to the following two reasons: first, the proportion of construction in OECD countries is much lower than that of manufacturing; second, the construction industry in OECD countries is relatively fully developed and it follows a sound standard, where such pollutants as construction dust are better controlled, so its impact on *PM$_{2.5}$* is not conspicuous. Electricity consumption has no distinct impact on *PM$_{2.5}$*. As we expected, the higher the proportion of environmental investment in public expenditure, the lower the average concentration *of PM$_{2.5}$*, demonstrating that environmental investment has a distinct emission reduction effect. In general, ENGOs have a perceptible negative impact on *PM$_{2.5,}$* which is not affected by the fixed effect of time and region, but may have an interaction with other control variables, thus weakening the environmental governance effect.

The ENGO's governance effect on *PM$_{2.5}$* is also reflected in its impact on the proportion of people exposed to an average concentration of *PM$_{2.5}$* above 10μg/m$^3$. As is shown in columns (5) and (6), the coefficients of ENGOs to *PM$_{2.5}$_pop* are significantly negative at 1% level, which are -1.955 and -2.967, respectively, indicating that for each additional unit of ENGOs, the proportion of the population exposed to an average concentration of *PM$_{2.5}$* above 10μg/m$^3$

will decrease by 2%-3% after controlling other influencing factors. For rest controllable variables, the growth rate of manufacturing and construction industry has no obvious impact on $PM_{2.5}\_pop$; while the share of manufacturing and construction in GDP has a marked positive effect on it; the influence of power generation on it is notably negative; and the proportion of environmental investment in public expenditure has no obvious impact on it. The main reason is that the proportion of population exposed to $PM_{2.5}$ above 10μg/m$^3$ is affected by the population density and economic development level of a country. For countries with the same $PM_{2.5}$ concentration level, their residents may be completely exposed to $PM_{2.5}$ above or below 10μg/m$^3$.

## 4.2. Carbon emission reduction effect of ENGOs

As the main contributor of greenhouse gases, $CO_2$ is the main cause of global warming. The environmental problems caused by global warming have an important destructive effect on the survival of human beings. Therefore, this part mainly discusses the relationship between ENGOs and $CO_2$. Table 3 reports the carbon emission reduction effects of ENGOs on carbon dioxide. The two models control the region fixed effects and their F-test values are high, which have passed the significance test. The $R^2$ of the two models are both greater than 0.6, an indicator of their strong explanatory power. The column (1) does not control the year fixed effects while column (2) controls the year fixed effects. The estimation coefficients of the *engo_pop* in

**Table 3. Carbon emission reduction effect of ENGOs.**

| Explanatory variable | Dependent variable: Ln$CO_2$ | |
|---|---|---|
| | **(3)** | **(4)** |
| *engo_pop* | -0.209*** | -0.212*** |
| | (0.021) | (0.021) |
| *Secondr* | -0.008 | -0.016 |
| | (0.008) | (0.011) |
| *Second* | -0.011 | -0.013* |
| | (0.007) | (0.007) |
| *Ele* | 0.012*** | 0.012*** |
| | (0.001) | (0.001) |
| *Env_I* | 0.150*** | 0.176*** |
| | (0.033) | (0.030) |
| *Buildr* | 0.003 | 0.000 |
| | (0.007) | (0.007) |
| *Build* | 0.044 | 0.042 |
| | (0.035) | (0.035) |
| Constant | 3.412*** | 3.632*** |
| | (0.253) | (0.303) |
| *N* | 510 | 510 |
| Year fixed effect | N | Y |
| Region fixed effect | Y | Y |
| $R^2$ | 0.604 | 0.608 |
| F | 73.066 | 27.665 |
| | [0.000] | [0.000] |

*Notes*: The numbers in parenthesis are robust standard errors; P values are in square bracket;

*, **, and *** represent 10%, 5%, and 1% significant levels, respectively.

two models are significantly negative. the ENGO's coefficients are -0.209 and -0.212 respectively, suggesting that when *engo_pop* increase by one unit, Carbon dioxide emissions will be reduced by about 21%, which means that the estimate results are relatively robust and credible. Hypothesis 1 is verified.

Among the control variables, the share of manufacturing in GDP has a major negative impact on $CO_2$, showing that the manufacturing in OECD countries has become a high-end industry and the manufacturing process does not produce too many $CO_2$. In addition, the growth rate of manufacturing, the proportion and growth rate of construction have no prominent impact on $CO_2$. The proportion of electricity production and environmental protection expenditure have a significant positive impact on $CO_2$, which may be linked to the fact that coal is still used to generate power, emitting a large amount of $CO_2$, and that environmental spending may not be used to address $CO_2$ but other pollutants that directly threaten human life, thus crowding out governance investment for $CO_2$, leading to a rise in greenhouse gas instead of decline.

### 4.3. *Oxynitride* emission reduction effect of ENGOs

Oxynitride includes a variety of compounds with different toxicity, such as $N_2O$, $NO$, $NO_2$, $N_2O_3$, $N_2O_4$ and $N_2O_5$, etc. In the air, however, most oxynitride are extremely unstable, and when exposed to light, humidity and heat, they become $NO$ and $NO_2$. Therefore, as for oxynitride, it is ENGOs' emission reduction effect on $NO$ and $NO_2$ that is mainly investigated.

Table 4 reports the estimated effects of oxynitride emission reduction by ENGOs. According to the estimation results of columns (1) and (2), whether the time fixed effect is controlled or not, ENGOs have a notable negative effect on *lnNO*. Their coefficients are -0.179 and -0.186 respectively, meaning that when *engo_pop* increases by a unit, nitric oxide will decrease by 18%~19%. Meanwhile, the estimated coefficients in columns (3) and (4) are significantly negative, being -0.179 and -0.186 respectively, indicating that when *engo_pop* increases by a unit, nitrogen dioxide will decrease by 26%~28%. Among the control variables, the coefficient of manufacturing is significantly negative, demonstrating that the manufacturing industry in those countries have developed into a more advanced stage which can reduce nitrogen oxide emissions. On the contrary, there is a conspicuous positive relationship between power generation and nitrogen oxide emissions. Besides, the higher the proportion of environmental investment, the more nitrogen oxide emissions will be. The higher the proportion of construction, the higher the nitrogen oxide emissions, which may because of the fact that some of the raw materials used in construction industry are heavy industrial products that produce a large amount of nitrogen oxides during the production process.

## 5. Robustness check

### 5.1. PM2.5 emission reduction effect of ENGOs

In the estimation above, we use the number of ENGOs owned by per million people as the core explanatory variable, yet its effect is different for countries with the same number of ENGOs but not the same size of population. Therefore, we use the number of ENGOs per ten thousand square kilometers as an alternative variable to test the robustness of the regression equation.

Table 5 shows the estimation result. Columns (1) and (2) consider the impact of *engo_sq* on $PM_{2.5}$ emission. The results show that the coefficients of ENGOs are significantly negative, signaling that the relationship between ENGOs and $PM_{2.5}$ emission is relatively stable. The coefficients, however, are relatively small, which is related to the measurement units of ENGOs. When *engo_sq* increases by a unit, the average concentration of $PM_{2.5}$ will decrease to 1.3μg/

**Table 4. Oxynitride emission reduction effect of ENGOs.**

| Explanatory variable | Dependent variable: Ln$NO$ | | Dependent variable: Ln$NO_2$ | |
| --- | --- | --- | --- | --- |
| | (1) | (2) | (3) | (4) |
| engo_pop | -0.179*** | -0.186*** | -0.262*** | -0.275*** |
| | (0.016) | (0.016) | (0.014) | (0.012) |
| Secondr | -0.013* | -0.026*** | 0.018* | 0.016 |
| | (0.008) | (0.009) | (0.010) | (0.011) |
| Second | -0.016*** | -0.021*** | -0.016* | -0.028*** |
| | (0.006) | (0.006) | (0.008) | (0.008) |
| Ele | 0.012*** | 0.012*** | 0.013*** | 0.012*** |
| | (0.001) | (0.001) | (0.001) | (0.001) |
| Env_I | 0.129*** | 0.187*** | 0.270*** | 0.390*** |
| | (0.042) | (0.034) | (0.047) | (0.038) |
| Buildr | 0.004 | -0.001 | 0.003 | -0.002 |
| | (0.006) | (0.006) | (0.007) | (0.007) |
| Build | 0.123*** | 0.117*** | 0.339*** | 0.324*** |
| | (0.033) | (0.033) | (0.038) | (0.038) |
| Constant | 5.231*** | 5.742*** | 3.038*** | 3.926*** |
| | (0.237) | (0.287) | (0.329) | (0.391) |
| N | 510 | 510 | 510 | 510 |
| Year fixed effect | N | Y | N | Y |
| Region fixed effect | Y | Y | Y | Y |
| R-squared | 0.601 | 0.622 | 0.570 | 0.608 |
| F test | 87.600 | 37.436 | 165.420 | 83.708 |
| (p-value) | [0.000] | [0.000] | [0.000] | [0.000] |

*Notes*: The numbers in parenthesis are robust standard errors; P values are in square bracket;

*, **, and *** represent 10%, 5%, and 1% significant levels, respectively.

$m^3$. Columns (3) and (4) show the effect of ENGOs per ten thousand square kilometers on the proportion of people exposed to $PM_{2.5}$ concentration above 10μg/m$^3$. The coefficients here are -1.404 and -1.104 respectively, which are extensively negative, indicating that when *engo_sq* increases by a unit, the proportion of population exposed to $PM_{2.5}$ higher than 10μg/m$^3$ will

**Table 5. PM2.5 emission reduction effect of ENGOs.**

| Explanatory variable | Dependent variables: $PM_{2.5}$ | | Dependent variables: $PM_{2.5}\_pop$ | |
| --- | --- | --- | --- | --- |
| | (1) | (2) | (3) | (4) |
| engo_sq | -2.03*** | -1.303*** | -1.404*** | -1.104*** |
| | (0.399) | (0.335) | (0.211) | (0.222) |
| N | 510 | 510 | 510 | 510 |
| Year fixed effect | N | Y | N | Y |
| Region fixed effect | Y | Y | Y | Y |
| Control variable | Y | Y | Y | Y |
| R-squared | 0.096 | 0.120 | 0.056 | 0.063 |
| F test | 5.630 | 2.402 | 26.594 | 8.374 |
| (p-value) | [0.000] | [0.000] | [0.000] | [0.000] |

*Notes*: The numbers in parenthesis are robust standard errors; P values are in square bracket;

*, **, and *** represent 10%, 5%, and 1% significant levels, respectively.

**Table 6. The mitigation effect of ENGOs on other air pollutants emission.**

| Explanatory variable | Dependent variables: $LnCO_2$ | Dependent variables: $LnNO$ | Dependent variables: $LnNO_2$ |
|---|---|---|---|
| | (1) | (2) | (3) |
| engo_sq | -0.066*** | -0.069*** | -0.140*** |
| | (0.005) | (0.005) | (0.005) |
| N | 510 | 510 | 510 |
| Year fixed effect | Y | Y | Y |
| Region fixed effect | Y | Y | Y |
| Control variables | Y | Y | Y |
| R-squared | 0.486 | 0.532 | 0.564 |
| F test | 50.282 | 43.090 | 128.071 |
| (p-value) | [0.000] | [0.000] | [0.000] |

*Notes*: The numbers in parenthesis are robust standard errors; P values are in square bracket;

*, **, and *** represent 10%, 5%, and 1% significant levels, respectively.

decrease by 1.1%-1.4%. Among the control variables, the proportion of manufacturing and construction has a positive impact on $PM_{2.5}$ emission; while power generation and environmental investment have a certain negative impact on $PM_{2.5}$, but it has no distinct impact on the proportion of population exposed to $PM_{2.5}$ above 10μg/m$^3$.

## 5.2. The mitigation effect of ENGOs on other air pollutants emission

Table 6 reports the estimation results of $CO_2$ and oxynitride. Column (1) lists the estimated result of carbon dioxide mitigation effect after controlling time and region fixed effect, with a coefficient of -0.066, passes the significance test at a level of 1%. Columns (2) and (3) consider the impact of *engo_sq* on nitric oxide ($LnNO$) and nitrogen dioxide ($LnNO_2$) respectively. The estimation results show that both coefficients are significantly negative, and there is narrowly no difference when comparing them with those from Table 4, implying that the estimation results of oxynitride emissions are also robust. Meanwhile, the estimated results of other controlled variable are basically consistent with the previous results.

## 6. Mechanism analysis

### 6.1. Econometric model of mechanism test

According to the theoretical mechanism diagram and theoretical hypothesis 2, we used the mediation effect model to test whether ENGOs would affect environmental quality through environmental investment. The specific model are as follows:

$$PM_{2.5_{it}} = \alpha_1 + c \times engo\_pop_{it} + Z \times \gamma_1 + \mu_i + \upsilon_t + \xi_{it} \tag{4}$$

$$Env\_I_{it} = \alpha_2 + a \times engo\_pop_{it} + Z \times \gamma_2 + \mu_i + \upsilon_t + \varsigma_{it} \tag{5}$$

$$PM_{2.5_{it}} = \alpha_3 + c\prime \times engo\_pop_{it} + b \times Env\_I_{it} + Z \times \gamma_3 + \mu_i + \upsilon_t + \zeta_{it} \tag{6}$$

$$c = c' + ab \tag{7}$$

In formula (4), the coefficient c represents the total effect of *engo_pop* on $PM_{2.5}$ emission. In formula (5), the coefficient *b* represents mediation effect of *engo_pop* on *Env_I*. In formula (6), the coefficient *b* represents the effect of the intermediate variable *Env_I* on the dependent

variable $PM_{2.5}$ after controlling the influence of *engo_pop*; the coefficient $c'$ is the direct effect of *engo_pop* on $PM_{2.5}$ after controlling the influence of *Env_I*. In formula (7), the coefficient *ab* represents the intermediate effect, which is also referred as indirect effect. The total effect is equal to the direct effect plus the indirect effect. $Z$ is the corresponding coefficient matrix of controlled variables, $\mu$ and $v$ represent the unobservable factors affected by region and time respectively, and $\xi_{it}, \varsigma_{it}, \zeta_{it}$ are random disturbance terms.

## 6.2. Mechanism test results

We used the external command *sgmediation* of Stata 16.0 to estimate the formula (4)–(7). Table 7 reports the estimation results. It can be found that *Sobel*, *goodman-1*, *goodman-2* and each coefficient. all passed the significance test at 5% level, a sign that the mediation effect model is reasonable. From the results of column (1), *engo_pop* has a major negative impact on $PM_{2.5}$ with a coefficient of -0.201. In column (2), the impact coefficient of *engo_pop* on *Env_I* is 0.05, which is exceptionally positive. In column (3), *engo_pop* still has a significant negative impact on $PM_{2.5}$, but the coefficient decreases to -0.176, and *Env_I* exerts a prominent negative impact on $PM_{2.5}$ with a coefficient of -0.5. Through the comparison of coefficients, it can be

**Table 7. Results of mediation effect model.**

| Explanatory variable | Dependent variable: $PM_{2.5}$ | Dependent variable: *Env_I* | Dependent variable: $PM_{2.5}$ |
|---|---|---|---|
| | (1) | (2) | (3) |
| *engo_pop* | -0.201*** | 0.050*** | -0.176** |
| | (0.076) | (0.006) | (0.080) |
| *Env_I* | | | -0.500* |
| | | | (0.269) |
| Constant | 10.338*** | -0.908*** | 9.884*** |
| | (1.831) | (0.236) | (1.917) |
| N | 510 | 510 | 510 |
| Time-fixed | Y | Y | Y |
| Regional-fixed | Y | Y | Y |
| Control variables | Y | Y | Y |
| R-squared | 0.116 | 0.184 | 0.121 |
| F test | 2.855 | 8.186 | 2.731 |
| (p-value) | [0.000] | [0.000] | [0.000] |
| Sobel-Goodman Mediation Tests | | | |
| | Coef | Z | P>Z |
| Sobel | -0.0286 | -1.912 | 0.055 |
| Goodman-1 | -0.0286 | -1.863 | 0.062 |
| Goodman-2 | -0.0286 | -1.966 | 0.049 |
| Coefficient a | 0.045 | 3.704 | 0.000 |
| Coefficient b | -0.628 | -2.233 | 0.025 |
| Direct effect | -0.028 | -1.912 | 0.056 |
| Indirect effect | -0.161 | -2.059 | 0.039 |
| Total effect | -0.190 | -2.447 | 0.014 |
| Proportion of total effect that is mediated: | | | 0.150 |
| Ratio of indirect to direct effect: | | | 0.177 |
| Ratio of total to direct effect: | | | 1.177 |

*Notes*: The numbers in parenthesis are robust standard errors; P values are in square bracket;

*, **, and *** represent 10%, 5%, and 1% significant levels, respectively.

found that when *engo* and *Env_I* are added at the same time, the coefficient of *engo_pop* on $PM_{2.5}$ decreases obviously, while the coefficient of *Env_I* on $PM_{2.5}$ is notably negative, which indicates that the effect of *Env_I* on $PM_{2.5}$ weakens the effect of *engo_pop* on $PM_{2.5}$. It also shows that *Env_I* does play a role of intermediate variable in the influence of *engo_pop* on $PM_{2.5}$.

In columns (1) and (3), the estimated results of each control variable are basically the same, and there is nearly no difference with the estimation results above. The results in column (2) show that higher proportion of manufacturing industry will promote investment in environmental protection, which can serve as an explanation to the policy of *control while polluting*. Nevertheless, the growth rate of manufacturing has a negative effect on environmental investment, suggesting that rapid growth of manufacturing industry will crowd out the investment in environmental protection. This is in line with the current situation of *treatment after pollution*. Moreover, power generation, the proportion and growth rate of construction industry all have a positive effect on environmental investment, which can also explain the policy of *control while polluting*.

## 7. Conclusions and policy implication

This paper selects relevant indicators of development level and environmental quality of ENGOs in OECD countries, investigates their effect on pollution mitigation and the mechanism they apply, and verifies that the existence of ENGOs can improve air quality in OECD countries. This paper concludes that first, the development of ENGOs in OECD countries can indeed reduce the average concentration of $PM_{2.5}$ and the proportion of people exposed to $PM_{2.5}$ above 10μg/ m$^3$ to a certain extent, and reduce $CO_2$ emission and overall emission of greenhouse gas (including $CO_2$, $NO_2$, $NO$ etc.). This conclusion is supported by consequential results that are measured by the number of ENGOs owned whether by one million people or ten thousand square kilometers. Second, through mechanism analysis, we verify that environmental investment plays an intermediate role between ENGOs and environmental quality.

Our empirical results encompass strong policy implications. As the growth of ENGOs improves environmental quality, a flourishing development of ENGOs and their environmental protection actions should be highly encouraged to curb polluting and reduce pollutants directly or indirectly. Given that environmental investment is essential for ENGOs in reducing pollution, more investment opportunities should be created for them by providing financial support, broadening investment channels or streamlining bureaucratic process.

## Author Contributions

**Conceptualization:** Guangqin Li.

**Data curation:** Qiao He, Dongmei Wang.

**Funding acquisition:** Bofan Liu.

**Methodology:** Qiao He, Bofan Liu.

**Software:** Dongmei Wang.

**Writing – original draft:** Guangqin Li, Qiao He, Dongmei Wang.

**Writing – review & editing:** Guangqin Li, Dongmei Wang, Bofan Liu.

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
