## [Decision Letter · Decision Letter 0]

12 Apr 2021

PONE-D-21-06791

Environmental Non-Governmental Organizations and Air-Pollution Governance: Empirical Evidence from OECD Countries

PLOS ONE

Dear Dr. Li,

Thank you for submitting your manuscript to PLOS ONE. After careful consideration, we feel that it has merit but does not fully meet PLOS ONE’s publication criteria as it currently stands. Therefore, we invite you to submit a revised version of the manuscript that addresses the points raised during the review process.

We look forward to receiving your revised manuscript.

Kind regards,

Ming Zhang, Ph.D.

Academic Editor

PLOS ONE

Journal Requirements:

[This work was supported by the Research Project of Anhui University of Finance and Economics (Green Development Effect of China’s Green Credit Policy)andby the Research Project of Sun Yat-sen University (Research on environmental protection goal setting and management effect of local government:Based on machine learning research method:13000-31610513)]

 [YES]

Reviewers' comments:

Reviewer's Responses to Questions

**Comments to the Author**

1. Is the manuscript technically sound, and do the data support the conclusions?

Reviewer #1: Yes

Reviewer #2: Yes

2. Has the statistical analysis been performed appropriately and rigorously? 

Reviewer #1: Yes

Reviewer #2: Yes

3. Have the authors made all data underlying the findings in their manuscript fully available?

Reviewer #1: Yes

Reviewer #2: Yes

4. Is the manuscript presented in an intelligible fashion and written in standard English?

Reviewer #1: Yes

Reviewer #2: Yes

5. Review Comments to the Author

Reviewer #1: This paper investigates the impact of ENGOs on air pollution emissions and finds that ENGOs have a positive impact on the improvement of environmental quality. The publication of this manuscript in “Plos One”is recommended after the following minor changes. Here are some comments for authors to modify.

1. The abstract should be modified, and it can not clearly describe the work of this paper.

2. It is better to modify “Theoretical hypothesis”,such as In Hypothesis 1, you put forward “With the growth of ENGOs, the environmental pollution will alleviate and environmental quality will be further ameliorated”. However, you did not analyze the relationship between the two above.

3. There are some mistakes in the paper, such as“the key variable in our analysis,;”in P6；“a sign that that the mediation effect model is reasonable”in P19.

Reviewer #2: This article mainly discusses the impact of environmental non-governmental organizations on air-pollution governance. The topic of the paper is relatively new, but there are still some problems that need improvement.

(1) 2.1 Literature reviews lacked logic, which only made the list of the relevant literatures simply without classification and summary.

(2) Hypothesis 1: The author listed the number of companies from the United Kingdom, the United States, Germany, and South Korea who have joined ENGOs, but there is no sufficient reason why these four countries are chosen.

(3) Hypothesis 2:

a. the author had no definition of informal regulation

b. inadequate explanation, such as: data, previous literatures

(4) 3.1. Model specification, the corresponding coefficient in the paper do not correspond to the equation (1)

(5) 3.2.2 Explained variables, Greenhouse-gas is composed of water vapor (H2O), carbon dioxide (CO2), nitrous oxide (N2O), freon, methane (CH4), etc. which overlaps with other air pollutants and is should to be deleted.

(6) 6.1. Econometric model of mechanism test

a. the coefficients should not be the same in equation (4) (5) (6)

b. Why to investigate whether ENGOs and PM2.5 have an interactive effect on residents’ health? What is the significance of exploring the interaction term of independent and dependent variable?

6. PLOS authors have the option to publish the peer review history of their article (what does this mean?). If published, this will include your full peer review and any attached files.

Reviewer #1: No

Reviewer #2: No

---

## [Author Response · Author response to Decision Letter 0]

11 May 2021

Response to Reviewers

First, we would like to thank reviewers for your insightful, constructive and helpful comments on our manuscript entitled “Environmental Non-Governmental Organizations and Air-Pollution Governance: Empirical Evidence from OECD Countries”. We have carefully considered and addressed all the comments and made necessary revisions in the revised manuscript. We provide a point-by-point response to the reviewers’ comments below.

The points raised by the reviewers are written in bold font, whereas our responses are shown in normal font, and the quotation of the revised manuscript is shown in italic font. 

Reviewer #1:

Reviewer #1: This paper investigates the impact of ENGOs on air pollution emissions and finds that ENGOs have a positive impact on the improvement of environmental quality. The publication of this manuscript in “Plos One” is recommended after the following minor changes. Here are some comments for authors to modify.

Response:

Thank you for taking time out of your busy schedule to review our paper. Thank you very much for your recognition of our work, too. In view of the shortcomings of the manuscript, we will continue to revise it in order to meet the publishing requirements.

1. The abstract should be modified, and it can not clearly describe the work of this paper.

Response:

Thank you very much for your kind, thoughtful and valuable comments. The abstract of the manuscript is not very good indeed. It is not very clear about the problems studied in the article through reading the abstract. In order to let readers know the content of the article directly, we have made the following modifications to the abstract in the revised manuscript:

…

Based on the panel data of environmental NGOs and air pollution in OECD countries, this paper uses econometric model to investigate the governance effect of environmental NGOs on air pollution. The results show that: ENGOs have a positive impact on the improvement of environmental quality, and the results are still valid after a series of robustness tests; Further mechanism analysis found that the environmental improvement by ENGOs is mainly achieved by increasing investment in environmental protection, which helps improve residents’ life expectancy and birth rate. This study provides empirical evidence for the effect of ENGOs on air pollution, and further provides ideas for environmental governance.

…

2. It is better to modify “Theoretical hypothesis”, such as In Hypothesis 1, you put forward “With the growth of ENGOs, the environmental pollution will alleviate and environmental quality will be further ameliorated”. However, you did not analyze the relationship between the two above.

Response:

Thank you very much for your kind, thoughtful and valuable comments. Hypothesis 1 is based on the facts that the environmental non-governmental organizations in several developed countries are developing and growing, as well as the environmental quality process of these countries. But this part of the content is not well written, we have made a comprehensive revision of this part of the content. The revised content are as follows:

…

What is the relationship between ENGOs and environmental quality? The following is an analysis of the development of ENGOs in the UK, the United States, Germany and South Korea, which have experienced environmental pollution problems in the process of industrialization. 

The United Kingdom, the first industrialized country, was most severely affected by environmental pollution, as evidenced by the smog incident in London in the 1950s [26]. ENGOs that rise with environmental pollution have developed rapidly in the UK, going through three stages characterized by elitism, populism and strategicism, and played the role of a third-party manager in making up for the failure of the state and the market [27]. After more than a century of development, the number of ENGOs registered in the UK has reached 200 according to Hilton et al. [28].

In the United States, ENGOs are a substantial environmental force outside all levels of government and scientific research institutions [29]. As a principal organ of environmental protection movements, they lodge lawsuits to court, appeal to parliament and lobby for specific environmental issues, and eventually realize management, compensation, supervision and control over pollution and ecological destruction through legislation. As the largest ENGO in the US, the World Wildlife Fund (WWF) has 1.2 million members across the country and international branches in dozens of countries, including China [30]. In addition to large organizations, there are also many small ENGOs in the US, such as the Pacific Environment and Resource Center established in 1987.

In the 1950s and 1960s, Germany was eager to emerge from the backwardness of its post-war situation and took a path of pollution before treatment, leading to increasingly severe environmental pollution, where industrial waste filled Rhine River and no blue sky could be seen at Ruhr industrial zone [31]. The ecological environment in Germany today, however, has been improved exceptionally, thanks to ENGOs’ immense contribution. Germany has over a thousand ENGOs, large and small, with more than two million employees. The largest one is the Nature Conservancy, with a history of over 100 years and about 400,000 members.

South Korea has also experienced serious pollution caused by rapid industrialization that has spawned a group of socially responsible ENGOs, of which the Environmental Movement Alliance is the largest, with branches and 85,000 members in 47 regions. Driven by the organization, South Korea's ENGOs have been growing and played a pivotal role in the environmental struggle from 1988 to 1992 mainly by participating in the environmental campaign against the construction of new nuclear power plants. 

Developed countries have experienced the stages from pre-industrialization to industrialization and then post-industrialization; therefore, their environmental governance has correspondingly gone through the stages of pollution before treatment, control while pollution and re-control with less pollution, which ENGOs have been growing in this process. It can be seen from such developed countries as the United Kingdom, the United States, Germany and South Korea that ENGOs have been associated with environmental pollution and become major participants and promoters of environmental pollution control. Therefore, we believe that there is a hypothetical relationship between the development of ENGOs in these countries and the environmental quality of these countries:

Hypothesis 1: With the growth of a country's environmental NGOs, the country's environmental pollution will be reduced (environmental quality will be improved).

…

3. There are some mistakes in the paper, such as“the key variable in our analysis,;”in P6；“a sign that that the mediation effect model is reasonable”in P19.

Response:

Thank you very much for your kind, thoughtful and valuable comments. Due to our carelessness and not careful examination of the manuscript, there are still some grammatical errors and language problems in the manuscript. We will revise it together in revised manuscript.

 

Reviewer #2: This article mainly discusses the impact of environmental non-governmental organizations on air-pollution governance. The topic of the paper is relatively new, but there are still some problems that need improvement.

Response:

Thank you for taking time out of your busy schedule to review our paper. Thank you very much for your recognition of our work, too. In view of the shortcomings of the manuscript, we will continue to revise it in order to meet the publishing requirements.

(1) 2.1 Literature reviews lacked logic, which only made the list of the relevant literatures simply without classification and summary.

Response:

Thank you very much for your kind, thoughtful and valuable comments. In the manuscript, there is no logic in the literature review part. In the revised draft, we classified the ways of environmental non-governmental organizations to participate in environmental governance, forming a relatively clear logical main line. The revised content are as follows:

…

To sum up, there are several ways for ENGOs to participate in Environmental Governance: first, ENGOs can raise their environmental awareness by holding environmental campaigns [15]; second, if their domestic ENGOs are relatively weak, they can improve their environmental discourse right by cooperating with international ENGOs in other developed countries [13]; third, in order to better participate in environmental governance, ENGOs take the initiative to establish interactive relations with enterprises and the government, so as to promote the realization of the normal functions of ENGOs [16, 17]; fourth, ENGOs cooperate with media organizations to transmit environmental information to the public through the media, and improve the public's environmental awareness [18]; fifth, for some countries with relatively small land area, environmental governance needs transnational cooperation, and ENGOs also need to cooperate closely with ENGOs in other countries to jointly deal with cross-border pollution problems [16, 19, 20, 21]; Sixth, ENGOs regularly publish reports on the environmental behavior of the government or enterprises to the public through information disclosure, so as to promote the government and enterprises to adopt more green environmental behavior [22, 23, 24, 25]. With the growth of ENGOs, the fields of different ENGOs are constantly subdivided, and the ways to ENGOs participate in environmental governance are constantly changing.

…

(2) Hypothesis 1: The author listed the number of companies from the United Kingdom, the United States, Germany, and South Korea who have joined ENGOs, but there is no sufficient reason why these four countries are chosen.

Response:

Thank you very much for your kind, thoughtful and valuable comments. Hypothesis 1 does not explain why the analysis of several countries leads to some abrupt content. In the revised draft, we added a paragraph.

…

What is the relationship between ENGOs and environmental quality? The following is an analysis of the development of ENGOs in the UK, the United States, Germany and South Korea, which have experienced environmental pollution problems in the process of industrialization. 

…

(3) Hypothesis 2:

a. the author had no definition of informal regulation.

b. inadequate explanation, such as: data, previous literatures

Response:

Thank you very much for your kind, thoughtful and valuable comments. In the revised draft, we define informal environmental regulation, and quote some literature to support our point of view. Other unreasonable aspects of hypothesis 2 are revised and some references are cited. According to the structure of the full text, we modify it as follows:

…

Environmental governance can be classified into formal and informal regulation [32, 33]. Formal environmental regulation mainly refers to the administrative and economics measures taken by a government [34]. The former includes implementing administrative orders to reduce pollution emissions or the use of chemical energy, restricting entry access for enterprises with certain industrial attributes, and shutting down pollution enterprises compulsorily. The effect of administrative measures is remarkable. The economic measures include taxation to pollutants, where the result is determined by tax severity and the ability of enterprises to bear the tax, or the provision of subsidy to nice and cleaner technology. Informal environmental regulation mainly refers to the way to restrict environmental behavior by improving residents' moral level and environmental awareness [35]. While formal environmental regulation addresses environmental issues directly, informal regulation exerts little immediate effect on environmental governance; instead, it achieves such effect through the influence of other factors to reduce environmental pollution and improve environmental quality. The role of ENGOs is informal regulation in environmental governance [25]. It can be seen from the previous analysis that ENGOs can influence environmental pollution. But how ENGOs affect environmental pollution is still unknown. Fig. 1 shows the main ways of how ENGOs influence environmental quality improvement.

Fig. 1. Impact mechanism of ENGOs on environmental quality improvement

As is shown in Fig.1, the traditional mechanism of environmental governance of ENGOs mainly consists of two parts. First, ENGOs ameliorate environmental quality mainly through education and information disclosure [19, 25]. As the publicity becoming more aware of environmental problems, they push the government to increase budget on environmental expenditure [7]. In addition, ENGOs meliorate environmental quality by influencing government environmental expenditures (including enterprises' environmental protection expenditure). However, due to data constraints, it is hard to obtain environmental awareness gained by residents through ENGOs’ education, but we can estimate the effect of environmental expenditure on ENGOs’ improvement of environmental quality. 

The fundamental starting point and goal of ENGOs participating in environmental governance is to improve the welfare level of residents [16]. With the improvement of environmental quality, residents' expectations for the future will also be raised, to increase their birth rate because children can live in a cleaner environment, and improve their life expectancy because they are free from various diseases caused by environmental pollution. Therefore, environmental NGOs can improve the environmental quality by improving the government's environmental investment, and the improvement of environmental quality will improve the birth rate of the whole society and the life expectancy of residents. Based on this, we propose the following hypothesis:

Hypothesis 2: ENGOs ameliorate environmental quality by means of increasing environmental expenditure, and further accelerate life expectancy and birth rate of the birth rate of the whole society.

….

(4) 3.1. Model specification, the corresponding coefficient in the paper do not correspond to the equation (1)

Response:

Thank you for your question. Because of our general intention, the formula description in the manuscript is inconsistent with the symbol in the formula. With your reminding, we have modified the formula and formula description. The revised content are as follows:

…

According to Hypothesis 1, we construct the following econometric model to investigate the impact of ENGOs on environmental pollution emissions:

 （1）

where subscripts and represent country and time respectively; Pollutant is the dependent variable, representing the emission of air pollution; engo is the key explanatory variable, representing the measurement index of ENGOs; Z is other control variables that affect the Pollutant; is the corresponding coefficient matrix of control variables; and are regional fixed effect and time fixed effect, respectively, indicating unobservable factors affected by region and time; is the random disturbance term; is the coefficient of our concern, where if it is less than zero, it shows that ENGOs have the emission reduction effect of environmental pollution, and the Hypothesis 1 can be verified.

…

(5) 3.2.2 Explained variables, Greenhouse-gas is composed of water vapor (H2O), carbon dioxide (CO2), nitrous oxide (N2O), freon, methane (CH4), etc. which overlaps with other air pollutants and is should to be deleted.

Response:

Thank you for your question. Indeed, when we wrote the manuscript, we did not notice the overlap between greenhouse gases and CO2. According to your suggestion, we decided to remove the greenhouse gas analysis and keep the other pollution gases. The reason is that carbon emission is the most concerned and important, while other nitrogen oxides represent the non-greenhouse gas pollutants, which also enriches the regression analysis of the paper. The contents of the revised manuscript are as follows:

…

As the main contributor of greenhouse gases, CO2 is the main cause of global warming. The environmental problems caused by global warming have an important destructive effect on the survival of human beings. Therefore, this part mainly discusses the relationship between ENGOs and CO2. Table 3 reports the carbon emission reduction effects of ENGOs on carbon dioxide. The two models control the region fixed effects and their F-test values are high, which have passed the significance test. The R2 of the two models are both greater than 0.6, an indicator of their strong explanatory power. The column (1) does not control the year fixed effects while column (2) controls the year fixed effects. The estimation coefficients of the engo_pop in two models are significantly negative. the ENGO’s coefficients are -0.209 and -0.212 respectively, suggesting that when engo_pop increase by one unit, Carbon dioxide emissions will be reduced by about 21%, which means that the estimate results are relatively robust and credible. Hypothesis 1 is verified. 

Table 3. Carbon emission reduction effect of ENGOs

Explanatory variable Dependent variable：LnCO2

 (3) (4)

engo_pop -0.209*** -0.212***

 (0.021) (0.021)

Secondr -0.008 -0.016

 (0.008) (0.011)

Second -0.011 -0.013*

 (0.007) (0.007)

Ele 0.012*** 0.012***

 (0.001) (0.001)

Env_I 0.150*** 0.176***

 (0.033) (0.030)

Buildr 0.003 0.000

 (0.007) (0.007)

Build 0.044 0.042

 (0.035) (0.035)

Constant 3.412*** 3.632***

 (0.253) (0.303)

N 510 510

Year fixed effect N Y

Region fixed effect Y Y

R2 0.604 0.608

F 73.066 27.665

 [0.000] [0.000]

Notes: The numbers in parenthesis are robust standard errors; P values are in square bracket; *, **, and *** represent 10%, 5%, and 1% significant levels, respectively.

…

(6) 6.1. Econometric model of mechanism test

a. the coefficients should not be the same in equation (4) (5) (6)

Response:

Thank you very much for your kind, thoughtful and valuable comments. In equation (4) (5) (6), each coefficient is different. In the manuscript, we only use different parameters to express the coefficients of the core variables, which is obviously not enough. In the revised version, we use different letters to represent different coefficients to be estimated.

…

According to the theoretical mechanism diagram and theoretical hypothesis 2, we used the mediation effect model to test whether ENGOs would affect environmental quality through environmental investment. The specific model are as follows:

 (4)

 (5)

 (6)

 (7)

In formula (4), the coefficient c represents the total effect of engo_pop on PM2.5 emission. In formula (5), the coefficient b represents mediation effect of engo_pop on Env_I. In formula (6), the coefficient b represents the effect of the intermediate variable Env_I on the dependent variable PM2.5 after controlling the influence of engo_pop; the coefficient is the direct effect of engo_pop on PM2.5 after controlling the influence of Env_I. In formula (7), the coefficient ab represents the intermediate effect, which is also referred as indirect effect. The total effect is equal to the direct effect plus the indirect effect. Z is the corresponding coefficient matrix of controlled variables, μ and v represent the unobservable factors affected by region and time respectively, and , , are random disturbance terms.

…

b. Why to investigate whether ENGOs and PM2.5 have an interactive effect on residents’ health? What is the significance of exploring the interaction term of independent and dependent variable?

Response:

Thank you very much for your question. Your question hit the nail on the head, and that's what we're struggling with. We also know that mechanism analysis is mainly to find the influence path of the core explanatory variables on the explained variables. At this point, we have found investment in environmental protection. However, that is why ENGOs want to promote the improvement of environmental quality. The ultimate goal is to return to the improvement of residents' well-being. We use the rising birth rate and life expectancy to measure residents' well-being. Therefore, we take this part as a part of mechanism analysis. We also feel that it is not appropriate to do so, which is not in line with the general norms of metrological thesis writing. If you feel that this part is a little redundant, we can also delete it.

---

## [Decision Letter · Decision Letter 1]

17 Jun 2021

PONE-D-21-06791R1

Environmental Non-Governmental Organizations and Air-Pollution Governance: Empirical Evidence from OECD Countries

PLOS ONE

Dear Dr. Li,

Thank you for submitting your manuscript to PLOS ONE. After careful consideration, we feel that it has merit but does not fully meet PLOS ONE’s publication criteria as it currently stands. Therefore, we invite you to submit a revised version of the manuscript that addresses the points raised during the review process.

We look forward to receiving your revised manuscript.

Kind regards,

Ming Zhang, Ph.D.

Academic Editor

PLOS ONE

Journal Requirements:

Reviewers' comments:

Reviewer's Responses to Questions

**Comments to the Author**

1. If the authors have adequately addressed your comments raised in a previous round of review and you feel that this manuscript is now acceptable for publication, you may indicate that here to bypass the “Comments to the Author” section, enter your conflict of interest statement in the “Confidential to Editor” section, and submit your "Accept" recommendation.

Reviewer #1: (No Response)

Reviewer #2: All comments have been addressed

2. Is the manuscript technically sound, and do the data support the conclusions?

Reviewer #1: (No Response)

Reviewer #2: Yes

3. Has the statistical analysis been performed appropriately and rigorously? 

Reviewer #1: (No Response)

Reviewer #2: Yes

4. Have the authors made all data underlying the findings in their manuscript fully available?

Reviewer #1: (No Response)

Reviewer #2: Yes

5. Is the manuscript presented in an intelligible fashion and written in standard English?

Reviewer #1: (No Response)

Reviewer #2: Yes

6. Review Comments to the Author

Reviewer #1: (No Response)

Reviewer #2: The authors have already done remarkable work on the revision. However, I still think it is not appropriate to investigate whether ENGOs and PM2.5 have an interactive effect on residents’ health, and I recommend deleting it.

7. PLOS authors have the option to publish the peer review history of their article (what does this mean?). If published, this will include your full peer review and any attached files.

Reviewer #1: No

Reviewer #2: No

---

## [Author Response · Author response to Decision Letter 1]

22 Jun 2021

Response to Reviewers

First, we would like to thank reviewers for your insightful, constructive and helpful comments on our manuscript entitled “Environmental Non-Governmental Organizations and Air-Pollution Governance: Empirical Evidence from OECD Countries”. We have carefully considered and addressed all the comments and made necessary revisions in the revised manuscript. We provide a point-by-point response to the reviewers’ comments below.

The points raised by the reviewers are written in bold font, whereas our responses are shown in normal font, and the quotation of the revised manuscript is shown in italic font. 

Reviewer #2:

Reviewer #2: The authors have already done remarkable work on the revision. However, I still think it is not appropriate to investigate whether ENGOs and PM2.5 have an interactive effect on residents’ health, and I recommend deleting it.

Response:

Thank you for taking time out of your busy schedule to review our paper again. I quite agree with your suggestion. In the revised draft, we delete this part of the content and modify the relevant content of the full text. I hope you will review our revised draft again. The specific revised parts are as follows:

Firstly, in the abstract, the result of residents’ life expectancy and birth rate is removed.

Secondly, in the hypothesis part, the hypothesis about life expectancy and birth rate is deleted;

Thirdly, in the description of variables and data, the description of variables and data about life expectancy and birth rate are deleted.

Fourth, in the part of mechanism analysis, we delete the empirical analysis on life expectancy and birth rate.

Fifthly, in the conclusions and policy recommendations section, we also deleted the conclusions and recommendations related to life expectancy and birth rate.

---

## [Decision Letter · Decision Letter 2]

12 Jul 2021

Environmental Non-Governmental Organizations and Air-Pollution Governance: Empirical Evidence from OECD Countries

PONE-D-21-06791R2

Dear Dr. Liu,

We’re pleased to inform you that your manuscript has been judged scientifically suitable for publication and will be formally accepted for publication once it meets all outstanding technical requirements.

Kind regards,

Ming Zhang, Ph.D.

Academic Editor

PLOS ONE

Additional Editor Comments (optional):

Reviewers' comments:

Reviewer's Responses to Questions

**Comments to the Author**

1. If the authors have adequately addressed your comments raised in a previous round of review and you feel that this manuscript is now acceptable for publication, you may indicate that here to bypass the “Comments to the Author” section, enter your conflict of interest statement in the “Confidential to Editor” section, and submit your "Accept" recommendation.

Reviewer #1: All comments have been addressed

Reviewer #2: All comments have been addressed

2. Is the manuscript technically sound, and do the data support the conclusions?

Reviewer #1: Yes

Reviewer #2: Yes

3. Has the statistical analysis been performed appropriately and rigorously? 

Reviewer #1: Yes

Reviewer #2: Yes

4. Have the authors made all data underlying the findings in their manuscript fully available?

Reviewer #1: Yes

Reviewer #2: Yes

5. Is the manuscript presented in an intelligible fashion and written in standard English?

Reviewer #1: Yes

Reviewer #2: Yes

6. Review Comments to the Author

Reviewer #1: Accept.This paper investigates the governance effect of ENGOs on air pollution in OECD countries, and this study is interesting and it can be publicated in PLOS ONE.

Reviewer #2: (No Response)

7. PLOS authors have the option to publish the peer review history of their article (what does this mean?). If published, this will include your full peer review and any attached files.

Reviewer #1: No

Reviewer #2: No

---

## [Editor Report · Acceptance letter]

3 Aug 2021

PONE-D-21-06791R2 

Environmental Non-Governmental Organizations and Air-Pollution Governance: Empirical Evidence from OECD Countries 

Dear Dr. Liu:

I'm pleased to inform you that your manuscript has been deemed suitable for publication in PLOS ONE. Congratulations! Your manuscript is now with our production department. 

Kind regards, 

on behalf of

Dr. Ming Zhang 

Academic Editor

PLOS ONE